

# Enhanced text clustering and sentiment analysis framework for online education: a BIF-DCN approach in computer education

Qingyun Zhang[1], Yang Li[1] and Muhammad Sheraz Arshad Malik[2]

[1] Office of Academic Affairs, Shijiazhuang Vocational College of Technology and Information, Shijiazhuang, Hebei, China
[2] Department of Software Engineering, Government College University, Faisalabad, Faisalabad, Pakistan

## ABSTRACT

Understanding students' emotional responses to course content and assignments is crucial for developing effective teaching strategies and improving online learning resources. To address this need, we propose a novel deep learning-based framework called BERT and BTF-IDF Integrated Framework with Deep Clustering Network (BIF-DCN), designed to accurately analyze student sentiment on educational platforms. The framework combines three key components: Bidirectional Encoder Representations from Transformers (BERT) for initial text feature extraction, Bi-level Term Frequency–Inverse Document Frequency (BTF-IDF) for enhanced feature representation, and an Improved Deep Embedded Clustering (IDEC) model for sentiment classification. BERT captures rich semantic features from student comments, which are further refined using BTF-IDF to highlight informative terms. These features are then clustered using the IDEC model to identify underlying sentiment-based topics. Experimental results show that BIF-DCN achieves higher clustering accuracy than existing IDEC-based and traditional single-model approaches on both public and self-constructed datasets. In addition to performance improvements, our method enables in-depth sentiment analysis of clustered topics, offering practical insights for optimizing teaching materials. This framework provides educators with valuable tools to better understand student needs and deliver more personalized and effective instruction, ultimately enhancing teaching quality and learner satisfaction.

# INTRODUCTION

The evolution of online education platforms has progressed from their early stages of technological experimentation to their current status as a ubiquitous form of education. Initially reliant on text-based materials and forum interactions during the infancy of Internet technology, these platforms laid the groundwork for distance learning. As technology advanced, the proliferation of broadband Internet and enhancements in multimedia technology substantially diversified the format and substance of online learning. Video lectures imbued distance education with vividness and intuitiveness, while the integration of

Corresponding author
Qingyun Zhang,
zhangxj@hebust.edu.cn

real-time interactive features fostered seamless communication between students and instructors (*Martin, Sun & Westine, 2020*). Moreover, the emergence of Massive Open Online Courses (MOOCs) heralded a new epoch in online education, vastly broadening the reach and accessibility of education through the provision of free or low-cost, high-quality course materials. In recent times, the integration of artificial intelligence and data analytics technologies has enabled online education platforms to deliver personalized learning experiences. This advancement not only enhances the efficiency and effectiveness of learning, but also optimizes learning trajectories and resource recommendations by analyzing students' learning patterns and performance (*Erlangga, 2022*).

Despite the significant strides and widespread adoption of online education platforms, the continual refinement of these platforms, both technically and content-wise, remains imperative for the ongoing optimization of pedagogy and the enhancement of educational quality. The sustained evolution and pedagogical enhancement of these platforms hinge crucially on a profound comprehension and adept application of student feedback. Serving as a direct and abundant source of feedback, student comments harbor invaluable insights into course content, instructional methodologies, assignment structuring, and platform functionalities (*Sun, Anbarasan & Praveen Kumar, 2021*). Consequently, through the astute analysis of topics and sentiments encapsulated in student comments, the platform can gain profound insights into students' learning journeys and requirements, thereby guiding the iterative enhancement of instructional content and platform functionalities. At the heart of this intelligent feedback system lies the utilization of deep learning, natural language processing, and allied technologies to autonomously discern and scrutinize pivotal topics and emotional inclinations within student feedback. This approach not only vastly enhances the efficiency and comprehensiveness of analysis, but also yields a more precise capture of nuanced emotional expressions and intricate topic interrelations compared to traditional manual analysis methods.

Through these sophisticated feedback analyses, educators can swiftly discern students' reception of specific instructional content, their feedback on assignment complexity, and their satisfaction with the platform experience. These insights serve as pivotal foundations for refining teaching methodologies and enhancing platform functionalities (*Qi et al., 2023*). Employing TextCNN, which leverages a convolutional neural network to capture local textual features, proves apt for the rapid processing of concise comment data. Meanwhile, the Bidirectional Encoder Representations from Transformers (BERT) model adeptly handles intricate textual relationships and delves into deep semantic nuances due to its pre-trained semantic prowess, gleaned from extensive text corpora. The amalgamation of these techniques enhances the efficiency and accuracy of sentiment extraction and thematic identification from students' online feedback (*Onan, 2021*).

Furthermore, topic modeling methodologies such as Latent Dirichlet Allocation (LDA) autonomously reveal potential topic distributions within voluminous text corpora, shedding light on salient concerns embedded within students' comments, including content satisfaction and feedback on assignment difficulty. Concurrently, sentiment analysis tools like VADER and TextBlob provide nuanced assessments of sentiment

intensity, which is crucial for deciphering students' nuanced emotional responses. Thus, through the synergy of deep learning methodologies, textual topics are clustered and scrutinized, laying the groundwork for subsequent sentiment classification and the construction of online resources. The specific contributions of this article are as follows:

1. To enhance the performance of text data representation, an enhanced Term Frequency–Inverse Document Frequency (TF-IDF) feature representation method, termed Bi-level Term Frequency–Inverse Document Frequency (BTF-IDF), is proposed, which integrates weight and scale factors into the current text feature representation.

2. The establishment of the BERT and BTF-IDF Integrated Framework with Deep Clustering Network (BIF-DCN) framework involves enhancing feature fusion within the conventional Improved Deep Embedded Clustering (IDEC) framework, thereby facilitating the analysis of textual topic features and validating the results using public datasets.

3. Following the completion of topic theme clustering within the internally curated text dataset, the subsequent sentiment analysis of topics is conducted using the classical Attention Mechanism-based Bi-Gated Recurrent Unit (ATT-Bi-GRU) model, furnishing insights for the subsequent development of teaching resources.

The rest of this article is organized as follows: "Related Works" introduces related works on topic clustering in text analysis; "Methodology" describes the establishment of BTF-IDF; The experiment results and analysis are presented in "Experiment Result and Analysis". The discussion is given in "Discussion" and the Conclusion is drawn at the end.

## RELATED WORKS

### Text topic clustering study

Text clustering involves a multifaceted and meticulous analysis process, incorporating various key technologies and theoretical underpinnings, such as clustering algorithms, text processing, and feature selection, among others. The efficacy of text clustering hinges on the interplay between each facet within clustering algorithms. Notably, *MacQueen (1967)* introduced the seminal K-Means algorithm, which partitions data into specified clusters and offers mathematical proof, rendering it the cornerstone of clustering algorithms. Additionally, *Ester et al. (1996)* developed the Density-Based Spatial Clustering of Applications with Noise (DBSCAN) algorithm based on principles of data distribution density. *Yu et al. (2011)* innovatively amalgamated kernel functions with cluster membership optimization to enhance the K-means algorithm's efficiency and mitigate time and space costs. *Xie, Girshick & Farhadi (2016)* advanced clustering allocation by utilizing self-encoders for feature extraction, proposing a deep-embedded clustering method. *Sinaga & Yang (2020)* explored the determination of cluster numbers within the K-means framework, presenting the UK-means algorithm, which obviates the need for pre-set cluster numbers. *Zhang et al. (2021)* proposed the Self-supervised Contrastive Clustering Learning (SCCL) clustering framework, which supports data enhancement through comparative learning perspectives. *Lu et al. (2013)* utilized expectation-maximization (EM) clustering to automatically identify health-related topics in an online

health community, categorizing discussions on lung cancer, breast cancer, and diabetes into semantic categories to assess the efficacy of topic detection. *Mehrotra et al. (2013)* employed diverse aggregation techniques in preprocessing Twitter tweets, consolidating data into "macros" to enrich content and construct improved LDA models. *Hua (2016)* applied an enhanced single-pass clustering algorithm to detect financial text topics, introducing the Term Frequency–Inverse Word Frequency–Inverse Document Frequency (TF-IWF-IDF) weight calculation method and the multi-topic centers algorithm, which comprehensively considers word position effects and employs multi-topic centers for clustering.

## Text sentiment analysis research

Sentiment analysis, a branch of natural language processing, involves extracting sentiment content from textual information and categorizing the text accordingly. This technique finds diverse applications in daily life, such as analyzing product feedback and identifying negative reviews on online platforms. *Sontayasara et al. (2021)* utilized a support vector machine algorithm to classify sentiments expressed by people planning to travel to Thailand *via* Twitter, extracting keywords indicative of corresponding emotions. *Polpinij & Ghose (2008)* introduced an ontology-based sentiment classification method to analyze online product review datasets using text classifiers and support vector machine (SVM) algorithms. *Mustofa & Prasetiyo (2021)* conducted sentiment analysis on Twitter discussions related to a specific topic, employing a combination of dictionary and naive Bayes classifiers. *Turney (2002)* employed unsupervised learning algorithms to categorize reviews into favorable and unfavorable categories, such as cars, banks, movies, and tourist destinations. *Pang (2002)* employed supervised learning algorithms, including naive Bayes, max entropy, and SVM, for sentiment classification of movie reviews. *Bagheri & Islam (2017)* analyzed sentiments expressed in people's comments on fake news, fashion, and politics on Twitter, categorizing comments as positive, negative, or neutral. In recent years, deep learning has emerged as a prominent research focus. *Pavlopoulos, Malakasiotis & Androutsopoulos (2017)* proposed a deep attention mechanism for user comment analysis, yielding superior results compared to traditional recurrent neural network (RNN) methods. *Rezaeinia et al. (2019)* introduced four methods, including lexicography, lexical tagging, positional algorithms, and word2Vec/Glove embeddings, to enhance pre-training accuracy in sentiment analysis.

The research above highlights the predominant focus of text clustering on conducting thematic analysis of large text datasets, extracting keywords, and clustering them to facilitate text classification to a certain extent. Concurrently, with the ongoing advancement of machine learning technologies in the realm of artificial intelligence, further sentiment analysis of text can be achieved by building upon clustering outcomes. In the context of sentiment analysis for online education platforms, short text analysis can initially establish word vectors using techniques such as word segmentation. Subsequently, relevant machine learning and deep learning algorithms can be employed to accomplish final sentiment classification. Hence, it is viable to conduct research on online teaching management utilizing such a framework.

## METHODOLOGY
### Short text weighted TF-IDF method and BTF-IDF

TF-IDF, an acronym for Term Frequency-Inverse Document Frequency, is a prevalent statistical technique used in text mining and information retrieval domains. It serves to gauge the significance of a word within a document set or a specific document within a *corpus* (*Artama, Sukajaya & Indrawan, 2020*). When a word exhibits high frequency within a text (high TF) and occurs infrequently across other texts (high IDF), it is deemed to possess strong category differentiation and is considered as significant for that text. Its primary representation is delineated by Eq. (1):

$$\text{vec}(d) = \sum_{t \in d} w_t \cdot \textit{tfidf}(t, d) \tag{1}$$

where w is the word vector weight, and tfidf(t, d) represents the word label of the corresponding word vector, where tf represents the forward word frequency and idf represents the inverse word frequency, the tfidf of a d vocabulary at the moment of t is calculated as shown in Eq. (2):

$$\textit{tfidf}(t, d) = \textit{tf}(t, d) * \textit{idf}(t, d). \tag{2}$$

In practical applications, the traditional TF-IDF method overlooks the distribution of feature words across categories. It solely considers the distribution of feature words across the entire *corpus*. Since this article necessitates conducting clustering on students' comments, and the number of categories in the *corpus* is unknown beforehand, we aim to expedite the clustering process by recalculating the weights based on the variance fluctuation of words. We hypothesize that when the variance between the frequency of a feature word in a short text and its frequency in other texts is larger, it indicates significant fluctuation in the *corpus*, implying non-uniform distribution. Hence, such words are presumed to possess discriminatory potential for categories. The formula measures the variance fluctuation of a word across a set of short texts (*Grootendorst, 2022*):

$$\tau = \frac{\sqrt{\sum_i^N \left( TF(w_i) - TF(w_{i+1}) \right)^2}}{N + 1} \tag{3}$$

where $TTF(w_i)$ denotes the term frequency of the word w in the i-th short text. N is the total number of short texts. $\tau$ captures the degree to which the frequency of a word fluctuates between adjacent documents. A $\tau$ high suggests that the word appears unevenly across the documents—it is frequent in some and rare in others—indicating potential discriminative power. Unlike traditional TF-IDF, this measure reflects the distributional instability of the word across the *corpus*. Consequently, we establish the corresponding weight adaptive factor, denoted as α, based on the magnitude of this volatility, as depicted in Eq. (4):

$$\omega(t, d) = \frac{TIF(t)}{\sqrt{\sum_{t \in d} [TIF(t)]^2}} (1 + \alpha). \tag{4}$$

Equation (4) computes the adjusted weight of term $t$ in document $d$. In Eq. (4) $TIF(t)$ represents a variant of traditional TF-IDF. The denominator normalizes the weights over all terms in document $d$. $\alpha$ is an adaptive weighting factor derived from the fluctuation measure $\tau$ in Eq. (3) and increases proportionally with $\tau$. The more a word fluctuates across documents (*i.e.*, higher $\tau$), the higher its weight becomes through $(1 + \alpha)$. This highlights context-sensitive, category-discriminative words that traditional TF-IDF might overlook.

After introducing the adaptive weighting factor $\alpha$, we construct the document representation by summing the word vectors weighted by their corresponding fluctuation-adjusted values. This is shown in Eq. (5). Here, $\omega(t, d, \alpha)$ combines both the normalized TIF and an adjustment based on the word's fluctuation across the *corpus*. A larger fluctuation leads to a higher $\alpha$, which increases the word's influence in the document vector.

$$\vec{d} = \sum_{t \in d} w_t \cdot \omega(t, d, \alpha). \tag{5}$$

To address variability in text, particularly in short texts such as student comments, introduce a scale penalty factor $\beta$ as shown in Eq. (6):

$$\beta = \frac{1}{\sqrt{|d|}}. \tag{6}$$

This factor is used to normalize the overall magnitude of the document vector. By incorporating $\beta$, the influence of text length on the resulting feature representation is effectively mitigated, preventing longer documents from dominating the feature space.

$$\text{vec}(\alpha, \beta) = \sum_{t \in d} w_t \cdot \omega(t, d, \alpha, \beta) \tag{7}$$

where $\omega(t, d, \alpha, \beta)$ is the text feature with two types of factor constraints. In this article, to illustrate and represent the feature more conveniently, we will introduce the BTF-IDF feature with weight and scale factors.

## IDEC clustering algorithm

Given the characteristics of the student topic discussion text, this article employs a sophisticated deep modeling approach for clustering analysis within the analytical model. The IDEC combines deep learning and clustering techniques, aiming to improve the performance of traditional clustering algorithms in handling complex data distributions. IDEC learns a low-dimensional representation of the data *via* a deep self-encoder and conducts clustering on this representation, thereby concurrently engaging in feature learning and clustering. This methodology proves particularly apt for high-dimensional data, facilitating the discovery of intricate and abstract data structures. The IDEC algorithm comprises two principal components: the deep autoencoder and the clustering layer (*Qu et al., 2020*). The deep autoencoder is used to learn a compressed representation of the input data, while the clustering layer is responsible for clustering on this compressed representation. In IDEC clustering algorithm soft assignment *i.e.*, by calculating data $x_i$

distance from the center of mass of a cluster $c_j$ and estimating the probability that the data will be assigned to the center of mass of a cluster. $p_{ij}$ The process is shown by Eq. (8):

$$p_{ij} = \frac{\left(1 + \|x_i - c_j\|^2/\gamma\right)^{-\frac{\gamma+1}{2}}}{\sum_{j'}\left(1 + \|x_i - c_{j'}\|^2/\gamma\right)^{-\frac{\gamma+1}{2}}}. \tag{8}$$

Meanwhile, the corresponding soft allocation defines the auxiliary target distribution. $d_{ij}$, which can be expressed by Eq. (9):

$$d_{ij} = \frac{p_{ij}^2/f_j}{\sum_{j'}\left(p_{ij'}^2/f_{j'}\right)} \tag{9}$$

where $f_j = \sum_i p_{ij}$ is the soft assignment of the data to the clusters $j$ probability, and the auxiliary target distribution can be viewed as a normalization process designed to improve prediction by normalizing the loss contribution of each center of mass. In the original depth-embedded clustering approach, the degree of matching between the soft assignment and the auxiliary target distribution is considered through the *KL* Scatter, which defines the clustering loss L as follows:

$$L = KL(D \| P) = \sum_i \sum_j d_{ij}\log\frac{d_{ij}}{p_{ij}}. \tag{10}$$

The IDEC clustering algorithm is different from the original depth-embedded clustering method in that it retains all the structure of the self-encoder and adds the reconstruction loss of the self-encoder to the loss, assuming that the encoding part of the self-encoder is $E_w$, the decoding part is $D_w$, then the reconstruction loss formula is shown in Eq. (11).

$$L_r = \sum_{i=1}^{n} \|x_i - D_w(E_w(x_i))\|_2^2. \tag{11}$$

Thus, in the IDEC clustering algorithm, total loss *i.e.*, the weighted sum of clustering loss and reconstruction loss, is shown in Eq. (12):

$$L_T = L_r + \lambda L \tag{12}$$

where $\lambda$ represents the loss weight factor. By continuously iterating the fused loss $L_T$. This updates the self-encoder parameter weights, the clustering cluster center of mass, and the auxiliary target distributions. IDEC can perform clustering while learning a compressed representation of the data, allowing the clustering process to take into account the intrinsic structure and complexity of the data. By jointly optimizing the reconstruction loss and clustering loss, IDEC can better preserve the local structure of the data while discovering the inherent clustering within it. Therefore, compared to traditional clustering algorithms, IDEC is more effective in handling high-dimensional data, which is particularly suitable for complex data types such as images and text. Therefore, the method is used in the process of establishing model clustering, and the overall framework will be described in detail in the following subsection. To enhance the stability and effectiveness of clustering,

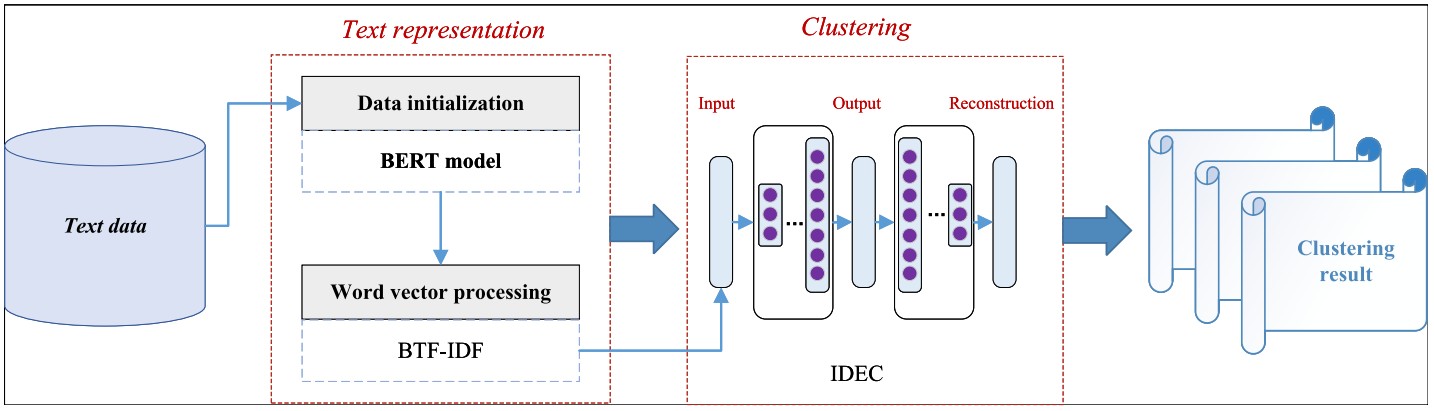

**Figure 1** The framework for the BIF-DCN.

the proposed IDEC-based framework adopts a two-stage training strategy. In the first stage, an autoencoder is trained in an unsupervised manner on the unlabeled student comment data to capture meaningful latent representations. Subsequently, the latent features extracted by the encoder are used to initialize cluster centroids *via* the K-means algorithm. In the second stage, joint optimization is performed by minimizing a combination of reconstruction loss and clustering loss. The target distribution for clustering is periodically updated to reflect the refined cluster assignments better. This initialization and training schedule ensures both robust representation learning and progressive refinement of clustering.

**Text clustering network based on BTF-IDF feature and IDEC (BIF-DCN)**

Following the completion of word vector feature extraction and the integration of weight and scale factors into the representation of text word vectors, clustering analysis based on the IDEC model is conducted. Subsequently, the corresponding output of clustering results is finalized. We designate this text clustering analysis framework as BIF-DCN, derived from feature naming and relevant acronym considerations. Figure 1 illustrates the simplicity of the form and the overall flow of the process.

The BIF-DCN framework proposed amalgamates the feature representation method with dual constraints introduced in "Experiment Setup" and the IDEC clustering framework. It comprises two components: the text representation part, denoted as BIF, and the clustering framework part, referred to as DCN. Initially, to enhance data quality, text preprocessing is conducted, utilizing the BERT model for initial feature extraction and the structured representation of textual data. In our framework, BERT is employed as a static feature extractor without fine-tuning. This decision is based on two key considerations. First, the size of the domain-specific dataset (student comments) is relatively small and imbalanced, which poses a risk of overfitting when fine-tuning large-scale pre-trained language models. Second, our goal is to build a lightweight and transferable pipeline that leverages BERT's general semantic representations while focusing on task-specific adaptation through the BTF-IDF weighting and IDEC clustering mechanisms. Although recent work shows that fine-tuned BERT often yields improved

results in classification tasks, our experiments demonstrate that combining static BERT embeddings with BTF-IDF reweighting achieves competitive and stable clustering performance, particularly in the short-text and low-resource setting of online education data. Future work may explore the integration of domain-adapted or fine-tuned transformers to enhance performance further.

Subsequently, the BTF-IDF method proposed is utilized for further feature extraction on the extracted vectors. Upon completion of feature extraction and representation, the IDEC framework is employed for clustering analysis of the model. During the clustering process, the data obtained undergo self-encoder training to initialize self-encoder parameters. Subsequently, joint tuning of the IDEC framework is performed, culminating in the final cluster analysis. Through the process above, it becomes evident that the proposed model synthesizes BTF-IDF features and the IDEC framework, thereby introducing corresponding innovation. Thus, it is named BIF-DCN. To ensure reproducibility and clarify the design of the IDEC module used in our framework, we provide detailed information on the encoder-decoder architecture and training parameters. The encoder consists of three fully connected layers with output dimensions of [500, 500, 2,000], followed by a latent embedding layer of size 10. The decoder is symmetric to the encoder. Rectified Linear Unit (ReLU) is used as the activation function for all hidden layers. The learning rate is set to 0.001, and the optimizer used is the Adam algorithm. In Eq. (12), the loss function combines the reconstruction loss and clustering loss, which are balanced by the parameter $\lambda$. In our experiments, we set $\lambda = 0.1$, following prior work, which achieves a good trade-off between preserving input structure and guiding cluster assignment. This configuration ensures stable training and high-quality latent feature learning for downstream clustering.

## Student sentiment analysis based on multiple attention mechanisms

Following the completion of text topic clustering analysis, our goal is to enhance the design of teaching resource classification by employing sentiment analysis of text comments using deep learning methods. In this article, we utilize the well-established Attention Mechanism-based Bi-GRU (ATT-Bi-GRU) framework (*Liu, Du & Zhou, 2020*). Below, we provide a succinct overview of the algorithm process, as illustrated in Algorithm ATT-Bi-GRU:

---

**Algorithm   ATT-Bi-GRU.**

**Input: Clustered data using BIF-DCN**

**Output:** Trained **ATT-Bi-GRU** model

**Initialization:** convert words to vectors, batch size, leaning rate, epochs, weights, bias.

**Define** Bi-GRU and multi-head attention mechanism

**Combine** attention module and Bi-GRU

**while** iteration < max-epoch **do**

    Feed the data to the ATT-Bi-GRU.

---

(Continued)

| Algorithm (continued) |
|---|
|    Weights and model update; |
|    If iteration==max epoch or loss satisfaction |
|    **End** |
| **Hyperparameter tuning** |
| **while** iteration < max-epoch **do** |
|    Feed validation data to ATT-Bi-GRU. |
|    Loss and gradients calculation. |
|    Model updated. |
|    **Compute and compare** loss |
|    **Save** ATT-Bi-GRU |
| **End** |

For the online teaching management resource classification design outlined in this article its primary objective is to spearhead the clustering of the principal topics within the model. Subsequently, leveraging this foundation, sentiment analysis of the content from more in-depth student discussions is undertaken. Only through this integrated approach can we achieve precise resource optimization and design. The integration of BERT, BTF-IDF, and IDEC within the BIF-DCN framework is driven by their complementary strengths in handling short-text sentiment clustering. While BERT provides robust contextualized embeddings at the word and sentence levels, it does not inherently emphasize term-level distributional variance, which may be critical for clustering tasks, especially in domain-specific, short, and sparse texts such as student comments. To address this, we introduce BTF-IDF as a lightweight yet practical feature enhancement layer that reweights BERT-derived features based on their statistical fluctuation across the *corpus*. This hybrid representation allows us to retain a deep semantic understanding of BERT while introducing discriminative weights that improve clustering separability. We chose not to directly fine-tune BERT within the IDEC pipeline for two reasons: (1) fine-tuning large pre-trained models like BERT on relatively small educational datasets poses overfitting risks and demands substantial computational resources, and (2) IDEC operates more effectively on fixed-dimensional, meaningful input representations rather than high-dimensional contextual embeddings alone. Although Sentence-BERT offers sentence-level embeddings suitable for clustering, it lacks the customized term-level weighting flexibility afforded by BTF-IDF, which we found particularly valuable in highlighting subtle but domain-relevant textual signals in our dataset. Thus, the proposed combination aims to balance semantic richness, statistical discriminability, and model stability in the clustering process.

## Justification of the proposed method
### Selection method
The selection of techniques in our proposed BIF-DCN framework was driven by the need for a highly accurate and scalable sentiment analysis system tailored for online educational

platforms. The methodology focuses on feature extraction, feature enhancement, and classification, ensuring a robust pipeline for analyzing student reviews. The following selection criteria were applied to choose the best-suited techniques:

1. Feature Extraction—BERT

   - BERT was selected for text representation due to its contextualized embedding capabilities, which enable a deep understanding of sentiment polarity within student reviews.
   - Compared to traditional natural language processing (NLP) models (such as TF-IDF and Word2Vec), BERT offers superior feature extraction by capturing semantic dependencies and contextual meanings.

2. Feature Enhancement—BTF-IDF

   - To refine and optimize textual feature representations extracted by BERT, we implemented a hybrid weighting technique combining BERT embeddings with TF-IDF.
   - This method was chosen to address sparsity issues in student feedback and highlight essential terms within the educational domain that impact sentiment classification.

3. Clustering & Classification—IDEC

   - IDEC was incorporated as it effectively combines deep learning-based representation learning with clustering algorithms, ensuring high-precision sentiment classification.
   - Unlike traditional clustering methods (*e.g.*, K-Means, DBSCAN), IDEC optimizes both feature representation and clustering structure, leading to improved accuracy in detecting latent sentiment patterns in student feedback.

### Implementation process

The BIF-DCN framework follows a systematic three-phase approach:

1. Feature Extraction with BERT

   - Preprocessing student reviews (tokenization, stop-word removal).
   - Applying BERT embeddings to extract deep contextualized text features.

2. Feature Representation Enhancement with BTF-IDF

   - Integrating BERT-generated features with a TF-IDF weighting mechanism to emphasize important sentiment-bearing terms.

3. Clustering and Sentiment Analysis with IDEC

   - Performing deep embedded clustering to classify sentiment-driven topics.
   - Evaluating clustering accuracy using performance metrics such as Adjusted Rand Index (ARI), Normalized Mutual Information (NMI), and clustering purity.

### *Why these methods?*

The combination of BERT, BTF-IDF, and IDEC ensures an efficient, context-aware, and high-accuracy sentiment analysis framework. The comparative analysis with existing single-model approaches demonstrated that BIF-DCN outperforms traditional sentiment analysis techniques in handling complex student emotions, thereby providing valuable insights to enhance online learning experiences and inform effective teaching strategies.

## EXPERIMENT RESULT AND ANALYSIS

### Experiment setup

In this article, two classic textual news datasets have been chosen for analysis in textual clustering studies. These datasets include the Reuters dataset and the Goodreads Quotes dataset. The specific details of each dataset are provided in Table 1.

Considering the subsequent practical application requirements, we selected five categories of data from the two datasets for analysis, each consisting of approximately 2,500 texts. To effectively demonstrate the performance of the proposed methodological framework in this article, we compare it with several classic and similar methods in the field. The main comparison methods include Kmeans, TF-IDF + IDEC, *Li, Cai & Wang (2020)*, *Ma & Zhang (2015)*, and *Guo et al. (2017)*. *Li, Cai & Wang*'s *(2020)* method improves upon the clustering framework by introducing two Kmeans frameworks after feeding the data. *Ma & Zhang (2015)* and *Guo et al.*'s *(2017)* methods focus on partial enhancements in feature processing, encompassing word vector extraction and improvements in feature representation.

After confirming the dataset and experimental comparison methods, we introduce the evaluation metrics utilized. In this article, we employ three types of metrics for analysis: accuracy (ACC), normalized mutual information (NMI), and adjusted rand index (ARI). The experimental environment of this article is outlined in Table 2.

The hyperparameter settings used in the ATT-Bi-GRU model are summarized in Table 3. These parameters were selected based on preliminary experiments to ensure stable training and effective convergence. Providing them with support enhances the reproducibility of the experimental results.

In addition to accuracy, we evaluated the training time and resource efficiency of the proposed BIF-DCN framework in comparison with a simple baseline model using TF-IDF features and K-Means clustering. Experiments were conducted on a standard machine with 32 GB RAM and an NVIDIA RTX 3080 GPU.

On the Reuters dataset, BIF-DCN required approximately 25 min for complete training, while TF-IDF+KMeans completed in under 3 min. For the Goodreads Quotes dataset, training BIF-DCN took around 30 min, compared to 4 min for the baseline. The peak GPU memory usage during BIF-DCN training was approximately 5.2 GB, whereas the baseline model operated entirely on the CPU with negligible memory demands.

### Clustering result analysis and comparison

After detailing the experimental setup, we proceed with a comparison of the proposed clustering methods, such as BIF-DCN. The results of this comparison are depicted in Figs. 2 and 3.

**Table 1 The information for the public dataset.**

| Name | Number of categories | Samples number |
|------|---------------------|----------------|
| RCV1 (Reuters *Corpus* Volume 1) | 103 | >800,000 |
| Goodreads quotes | 27 | >80,000 |

**Table 2 The experiment environment.**

| Environment | Information |
|-------------|-------------|
| CPU | Intel i7-13700K |
| GPUs | RTX 3080 |
| Language | Python 3.10 |
| Framework | Pytorch |

**Table 3 The hyperparameters setting for the model.**

| Hyperparameter | Value |
|----------------|-------|
| Batch size | 64 |
| Learning rate | 0.001 |
| Number of epochs | 20 |
| Optimizer | Adam |
| Hidden size (GRU) | 128 |
| Attention mechanism | Additive attention |
| Dropout rate | 0.5 |
| Embedding dimension | 300 |

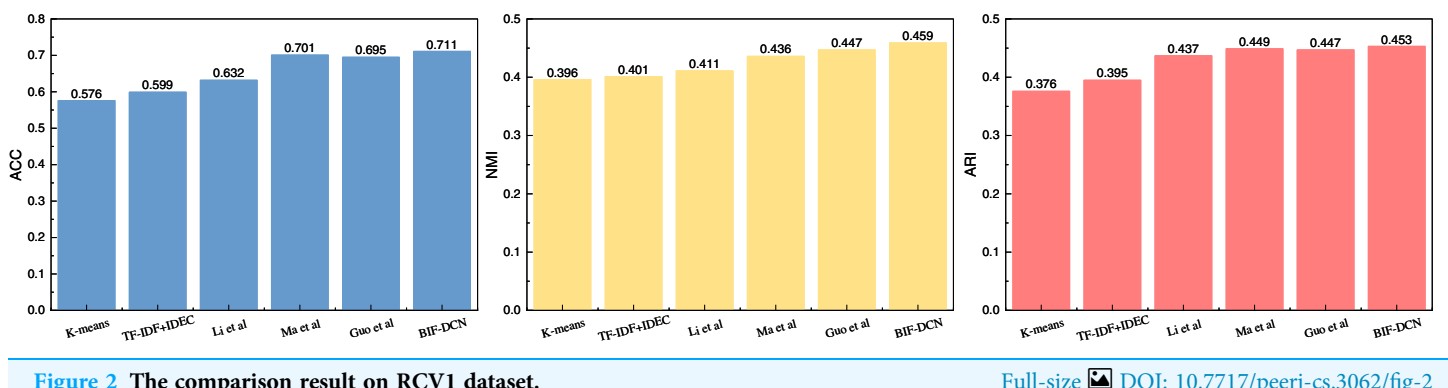

**Figure 2 The comparison result on RCV1 dataset.**

Figure 2 presents the results of different clustering methods applied to the RCV1 dataset across three evaluation metrics. From the data depicted in the figure, it is evident that the BIF-DCN method proposed outperforms other methods across all three indicators: accuracy (ACC), normalized mutual information (NMI), and adjusted mutual information (ARI). Notably, BIF-DCN exhibits a significant advantage in the NMI indicator, surpassing other methods by nearly 0.01. Additionally, it demonstrates superior

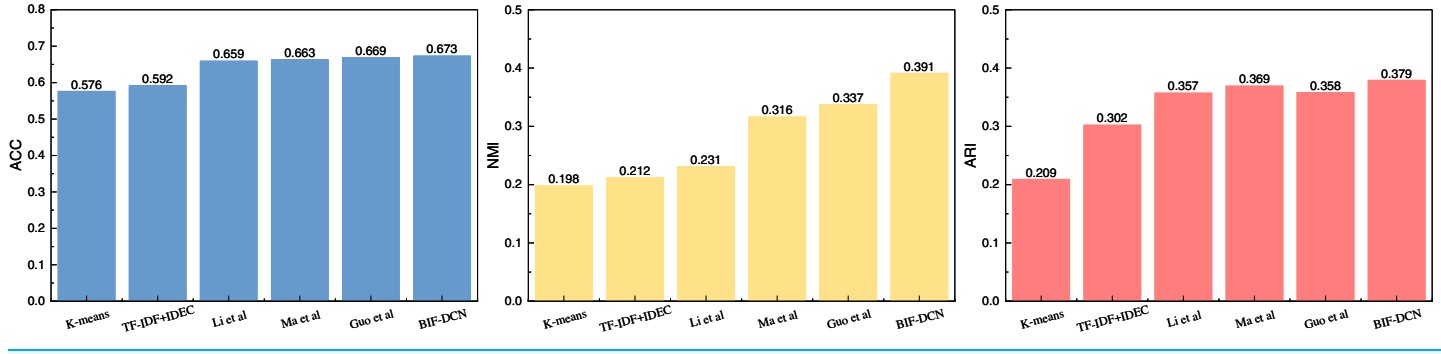

**Figure 3  The comparison result on Goodreads quotes dataset.**

performance across the remaining indicators as well. The clustering results for the Goodreads Quotes dataset are illustrated in Fig. 3.

In Fig. 3, it is apparent that due to the heightened complexity of the dataset utilized, the overall clustering performance is slightly inferior compared to the results obtained with the RCV1 dataset. Particularly, the advantage of the ACC index for the BIF-DCN method proposed is not as pronounced, exhibiting a marginally higher score compared to the other top-performing methods by approximately 0.04. However, the results under the NMI and ARI indices are relatively more prominent, with BIF-DCN leading by more than 0.01 in both cases.

To facilitate a more comprehensive comparison of the proposed methods across different feature representations, we conducted ablation experiments on the models during testing. Specifically, these experiments involved removing the weighting factor and scale factor added in this study. The results of these ablation experiments are depicted in Fig. 4.

In Fig. 4, the left data represents the results of the ablation experiments conducted on the RCV1 dataset, while the correct data pertain to the results obtained from the Goodreads Quotes dataset. Based on the ablation experiments, it is evident that the ACC metrics of the proposed method significantly outperform those of the method without the introduction of weight factor and scale factor metrics. Furthermore, the performance is also observed to be more stable across the test data. Moreover, the performance of the scale factor in the figure illustrates that it has a substantial impact on the overall stability performance of the model, yielding better improvements in model performance.

## Emotion analysis for the students' topic

To support the application of clustering methods in real-world educational settings, we constructed a domain-specific dataset based on student discussions from an online teaching platform. This self-collected dataset was built over five years and comprises a total of 7,532 anonymized comments from approximately 1,200 students, spanning multiple academic terms and course types. Following the structure and labeling protocol used in publicly available datasets, we manually annotated the comments into three major thematic categories: course content, teaching situations, and homework-related issues. The annotation process was conducted by a team of three educational researchers with domain

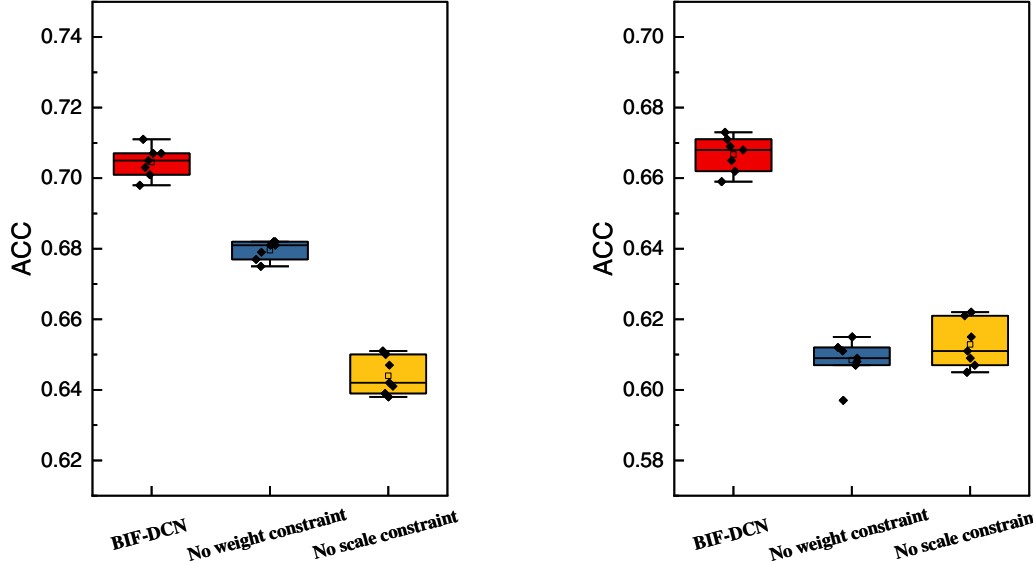

**Figure 4** **The ablation experiment result concerning ACC on RCV1 and Goodreads quotes dataset.**

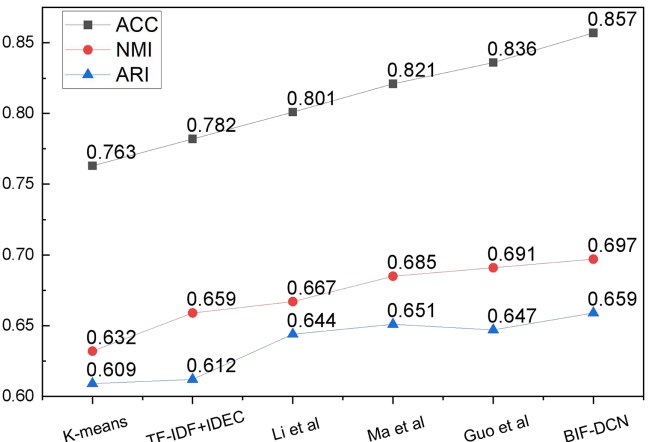

**Figure 5** **The clustering result on the established dataset.**

expertise. To ensure reliability, each comment was independently labeled by at least two annotators, with disagreements resolved through discussion or majority voting. Inter-annotator agreement, measured using Cohen's kappa, reached 0.82, indicating substantial consistency. In addition to thematic categorization, emotional tendencies were also analyzed within each cluster using a separate sentiment tagging schema. This emotional layer supported the downstream design of personalized educational interventions and the classification of teaching resources. To evaluate the performance and generalizability of the clustering methods, we conducted parallel experiments using both the public dataset and the self-constructed dataset. The latter serves as a critical testbed for exploring student behavior and feedback patterns in online learning environments.
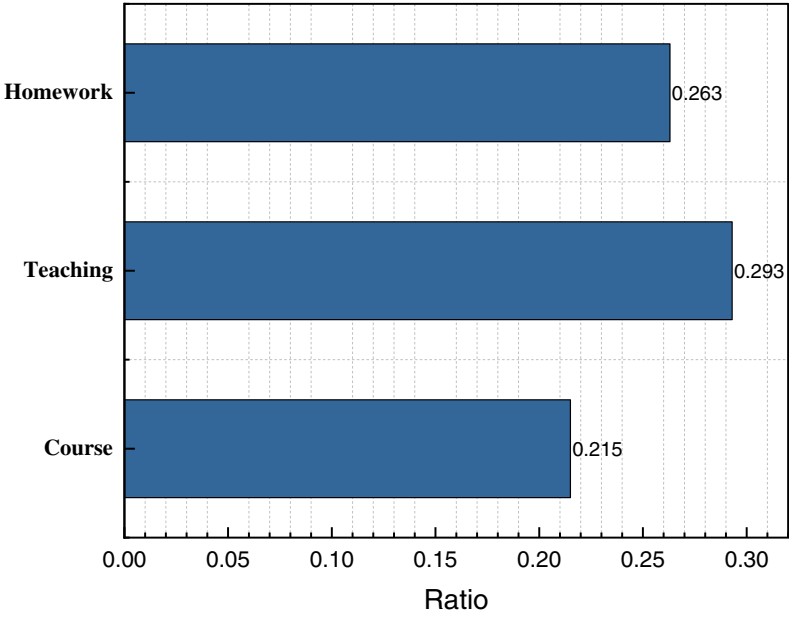

**Figure 6 The three topics with the highest ratio.**

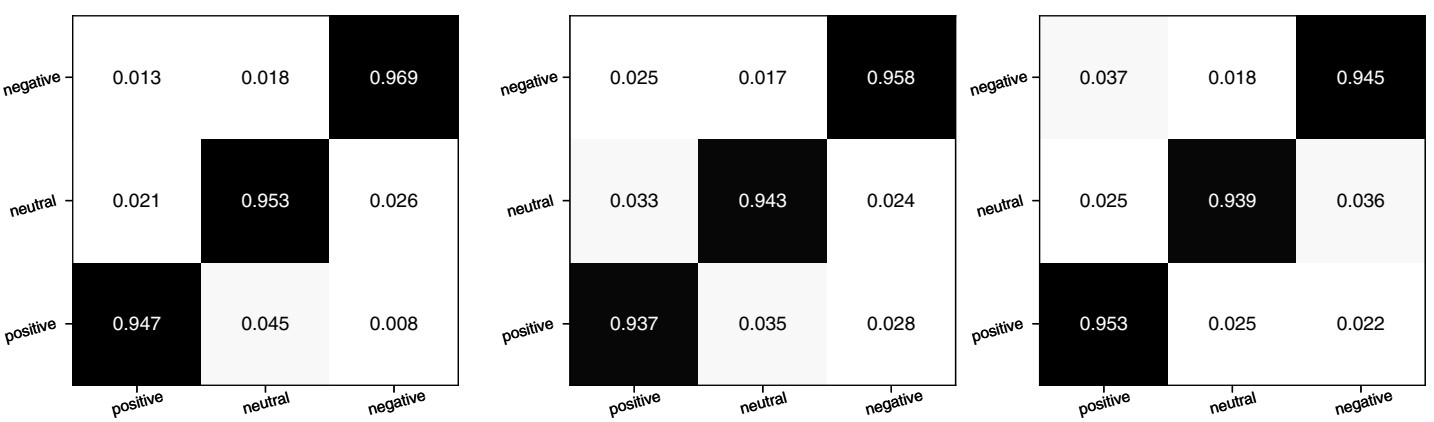

**Figure 7 The confusion matrix for the emotion classification concerning three topics.**

The results of this analysis are illustrated in Fig. 5.

Based on the trend of the curve depicted in Fig. 6, it is evident that the model proposed in this article exhibits commendable performance across the three indicators of accuracy (ACC), normalized mutual information (NMI), and adjusted rand index (ARI). Notably, its clustering ACC reaches 85.7%, which signifies outstanding performance in unsupervised analysis. This provides robust algorithmic support for subsequent analyses. Upon obtaining the corresponding clustered data, we proceed to analyze the top three themes—homework, teaching, and course—with proportions of 0.263, 0.293, and 0.215, respectively. Subsequently, following confirmation of the proportions of the respective models, we employ the relevant methods introduced in "Student Sentiment Analysis Based

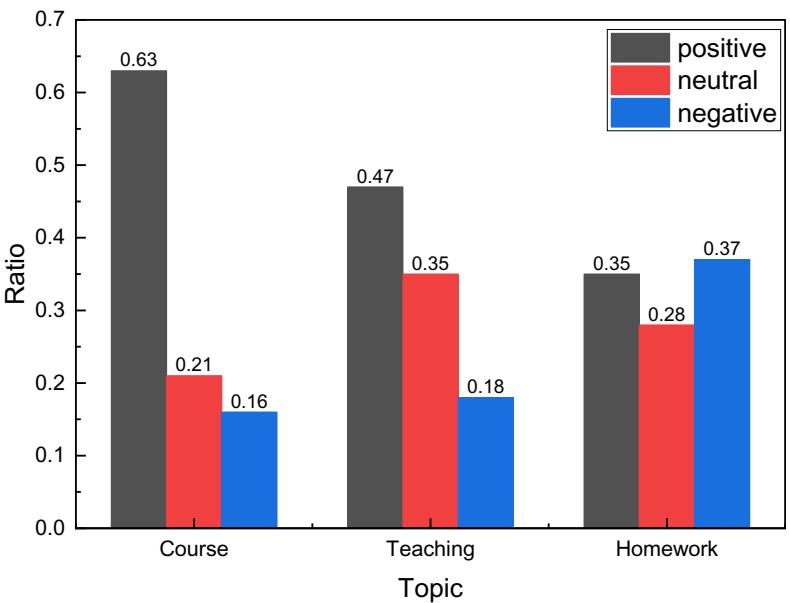

**Figure 8  The emotion ratio distribution.**

on Multiple Attention Mechanisms" to complete the sentiment classification of these data. The results of this test are depicted in Fig. 7.

In Fig. 7, the three plots represent the confusion matrices for sentiment classification of course, teaching, and homework data, respectively. By examining the color shades in the matrix, it becomes evident that the ATT-Bi-GRU method utilized exhibits higher accuracy in classifying the sentiment of much shorter texts. Consequently, this method can be effectively employed for further analysis of student topic sentiments. The overall recognition performance of the model for the three categories of emotions appears to be well-balanced, with all of them maintained at a level near 0.95. This suggests that even in cases of misclassification, these data have minimal impact on the sentiment classification of the topic. Furthermore, we conducted a detailed analysis of the different emotion percentages of the three types of topics, as depicted in Fig. 8.

The examination of the proportionate sentiment across various topics in Fig. 8 reveals notable trends. Specifically, regarding homework, neutral and negative sentiments are prevalent, outnumbering positive sentiments. This suggests a predominance of negative sentiments among students in this domain, necessitating further refinement of homework parameters. Conversely, within the domains of curriculum and teaching situations, sentiment analysis indicates a favorable outcome, characterized by a higher occurrence of positive sentiment and neutralizing procedures. This underscores the need for meticulous upkeep of these resources to sustain such encouraging outcomes.

To further evaluate the effectiveness of the proposed BTF-IDF feature representation, we conducted an ablation experiment comparing it with the traditional TF-IDF approach. In this experiment, we replaced the BTF-IDF module in our framework with standard TF-IDF, referring to the resulting model as TIF-DCN, while the original model is denoted

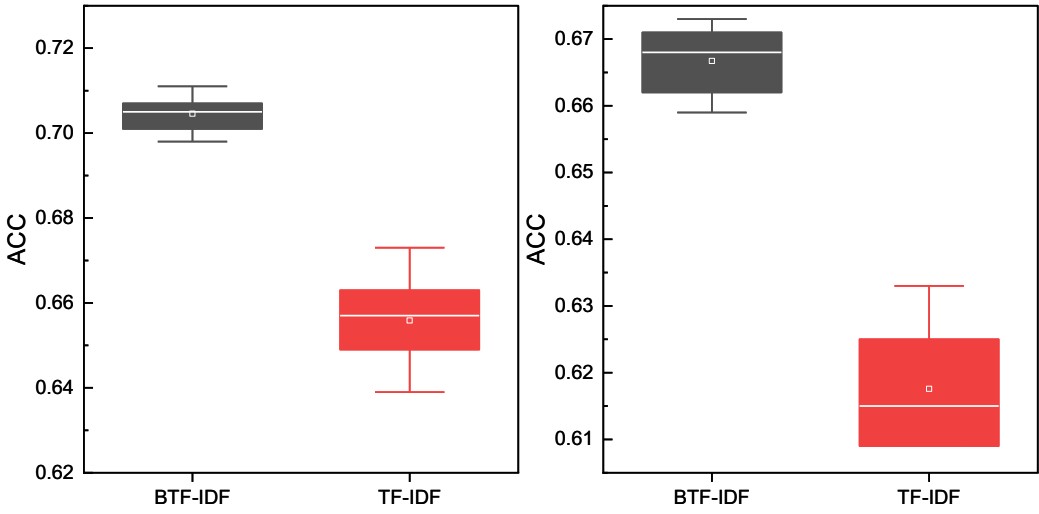

**Figure 9 The Ablation experiment result for the clustering results on both datasets between TIF-DCN and BIF-DCN.**

as BIF-DCN. Both models were evaluated on two datasets across three types of student sentiment scenarios.

As illustrated in Fig. 9, BIF-DCN consistently achieves higher clustering accuracy (ACC) than TIF-DCN. The performance gap is particularly pronounced in Goodreads Quotes, where TIF-DCN exhibits significant variability and reduced effectiveness. This result highlights the advantage of incorporating the variance-based weighting factor α and the scale penalty β in BTF-IDF.

## DISCUSSION

This article introduces the BIF-DCN model, tailored for analyzing topics within online platforms dedicated to student learning. The model combines the deep feature extraction capabilities of the BERT model with the efficient cluster analysis features inherent in the IDEC framework. Furthermore, it enhances textual representation by integrating the BTF-IDF method. By incorporating weighting factors, the model discerns and amplifies the significance of keywords or phrases pivotal for grasping the central idea and emotional tenor of student comments. This nuanced approach facilitates the more precise grouping of comments that share common themes during clustering, thereby enhancing clustering accuracy. Moreover, the inclusion of a scale factor adjusts the relative importance among different features, rendering the model adaptable to the diversity and intricacy of textual data. Such adaptability proves crucial in handling unstructured text data, such as student comments, which often encompass a wide range of topics and sentiment expressions. The scale factor enables the model to comprehend and categorize this multifaceted information more effectively. The BIF-DCN model effectively integrates deep feature representation with the clustering process through the IDEC framework, resulting in improved clustering efficiency.

Additionally, it refines feature representation iteratively during clustering, culminating in more precise and meaningful clustering outcomes. In contrast to traditional frameworks such as K-means and IDEC, the BIF-DCN model provides a richer and more detailed feature representation, thereby enabling a deeper semantic understanding and finer differentiation of semantic nuances. Consequently, the model exhibits superior performance in clustering both public datasets and self-constructed data, providing novel insights and methodologies for addressing future clustering challenges.

Utilizing the BERT model for initial text feature extraction and complementing it with the BTF-IDF method, the BIF-DCN approach demonstrates notable improvements in processing complex text data. Particularly in clustering analyses of student feedback on online platforms, this method adeptly captures profound semantic insights and subtle affective nuances, offering invaluable insights for comprehending student responses and refining educational content and methodologies. In handling unstructured text data, such as student comments, BIF-DCN excels at delineating key themes and nuanced affective differentiation through meticulous feature representations, enabling educational institutions to delve deeply into students' needs and optimize course designs. In this study, we expanded upon this potential by conducting sentiment analysis after topic clustering. By scrutinizing students' emotional reactions to specific educational content or resources, platforms can precisely match and recommend personalized learning materials, thus bolstering learning outcomes and user satisfaction. These sentiment analysis findings empower education providers to discern popular course content and areas requiring improvement, guiding the refinement and updating of courses to ensure their quality and relevance. Moreover, students' feedback on teaching methodologies can steer educators or platforms toward enhancing their teaching strategies and adopting more effective methods. Illustratively, the identification results presented herein demonstrate how platform assignment settings can be refined following sentiment analysis. Assignments can be structured hierarchically based on students' proficiency levels, ensuring each student encounters an appropriately challenging learning experience. Involving students in the design process of assignments fosters a sense of ownership over their learning journey and enhances their engagement with the assignments. Additionally, organizing stress management workshops to impart time management skills and relaxation techniques equips students with tools to better navigate academic pressures. By amalgamating cluster analysis and sentiment analysis outcomes, education platforms gain deeper insights into students' preferences and needs, enabling them to make informed decisions regarding resource development. Nonetheless, implementing this approach necessitates considerations regarding data processing accuracy, privacy safeguards, and the complexity of result interpretation.

## CONCLUSION

This study introduces a pioneering deep learning-based analysis framework, BIF-DCN, which meticulously conducts clustering and sentiment analysis of student feedback on online education platforms. By seamlessly integrating the BERT model, enhanced IDEC,

and innovative BTF-IDF method, the framework achieves precise recognition and analysis of student sentiment and feedback. The research attains clustering accuracies exceeding 0.65 on public datasets and surpassing 0.8 on self-constructed datasets, significantly outperforming traditional clustering and single deep learning model methods. Moreover, examining students' emotional attitudes after clustering provides an effective assessment method for discerning students' emotional expressions across various topics, thereby aiding educators in evaluating students' learning status on online teaching platforms. The development and optimization of educational resources within this framework offer novel perspectives and methodologies, thereby broadening the application scope of deep learning in education.

However, despite the notable advancements achieved using the BIF-DCN framework for sentiment and theme analysis of student comments on online education platforms, certain limitations are inevitable. Specifically, the model occasionally misidentifies comments with implicit or intricate sentiment expressions, erroneously categorizing negative or neutral sentiments as positive. This issue may arise from dataset linguistic ambiguity, insufficient contextual understanding by the model, or inadequate handling of polysemous vocabulary associated with particular sentiment expressions. Such challenges are prominent when students employ implicit or pun-laden language to express emotions, eluding accurate capture of their true emotional inclinations. Additionally, the study's effectiveness is constrained by the quantity and quality of the data, especially in relatively small or unbalanced datasets, which may limit model accuracy and generalization. Furthermore, the complexity of the sentiment analysis model poses a challenge, particularly in addressing diverse and intricate sentiment expressions, necessitating heightened model flexibility and adaptability. Addressing these challenges constitutes pivotal directions for future enhancements.

### Funding
The authors received no funding for this work.

### Competing Interests
The authors declare that they have no competing interests.

### Author Contributions
- Qingyun Zhang conceived and designed the experiments, performed the experiments, performed the computation work, prepared figures and/or tables, authored or reviewed drafts of the article, and approved the final draft.
- Yang Li performed the experiments, analyzed the data, performed the computation work, prepared figures and/or tables, authored or reviewed drafts of the article, and approved the final draft.
- Muhammad Sheraz Arshad Malik performed the experiments, analyzed the data, prepared figures and/or tables, and approved the final draft.

## Data Availability

The code is available in the Supplemental File.

The Reuters-MDV Dataset is available at Kaggle:

https://www.kaggle.com/datasets/nltkdata/reuters.

The Goodreads Quotes Dataset is available at Kaggle: DWS Studio. (2023). Goodreads Quotes Dataset [Data set]. Kaggle. https://doi.org/10.34740/KAGGLE/DSV/6605524.

## Supplemental Information

Supplemental information for this article can be found online at http://dx.doi.org/10.7717/peerj-cs.3062#supplemental-information.

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
