# Peer review of "Enhanced text clustering and sentiment analysis framework for online education: a BIF-DCN approach in computer education"

_PeerJ Computer Science, doi:10.7717/peerj-cs.3062_

## Round 0.1 · original submission · Major Revisions

Please see both reviewers' detailed comments. The reviewers acknowledge the paper's novel BIF-DCN framework for text clustering and sentiment analysis in online education, highlighting its structured presentation and timely relevance. However, they raise concerns about insufficient methodological clarity, particularly regarding the mathematical formulation of BTF-IDF, hyperparameter details (e.g., ATT-Bi-GRU, IDEC initialization), and the rationale for combining BERT, BTF-IDF, and IDEC. Both reviewers stress the need for more rigorous experimental validation, including ablation studies (e.g., TF-IDF vs. BTF-IDF), dataset descriptions (e.g., size, annotation), and implementation specifics (e.g., encoder-decoder architecture, fine-tuning BERT). Reproducibility issues are also noted, such as missing training schedules, optimization details, and computational efficiency. Suggestions include simplifying language, adding visualizations, and addressing typographical errors.

**Language Note:** The review process has identified that the English language must be improved. PeerJ can provide language editing services - please contact us at [email protected] for pricing (be sure to provide your manuscript number and title). Alternatively, you should make your own arrangements to improve the language quality and provide details in your response letter. – PeerJ Staff

·

Basic reporting

This paper presents a novel deep learning-based framework, BIF-DCN, for enhanced text clustering and sentiment analysis in online education platforms. By integrating BERT-based feature extraction, a modified BTF-IDF feature representation, and an improved IDEC clustering mechanism, the authors aim to better capture student sentiments and learning needs. The manuscript is well-structured, with a clear abstract, introduction, related works, methodology, experimental results, discussion, and conclusion sections.

Experimental design

- The BTF-IDF method, though innovative, needs clearer mathematical formulation. Some of the equations (like Eq. (3)–(6)) are difficult to follow and not sufficiently explained.
- The model hyperparameters for ATT-Bi-GRU are not explicitly reported (batch size, learning rate, number of epochs, optimizer). Including these would help in reproducibility.
- The description of the IDEC extension (Section 3.2) lacks clear information on initialization and training schedules.

Validity of the findings

The self-collected dataset is critical to the application domain but is only lightly described (lines 391–393). More details (e.g., number of students, number of comments, annotation process) are necessary for assessing dataset validity.

Additional comments

Simplify the language, particularly in the Abstract and Introduction, for broader accessibility.
Carefully check for typographical errors throughout the manuscript.

Reviewer 2 ·

Basic reporting

The paper proposes a hybrid deep learning framework (BIF-DCN) integrating BERT, a novel BTF-IDF representation, and IDEC clustering, followed by ATT-Bi-GRU for sentiment classification of clustered topics. The manuscript addresses a relevant and timely problem, but several aspects related to model design, theoretical justification, and implementation details require clarification and strengthening.

Experimental design

Following are the comments:

While the integration of BERT, BTF-IDF, and IDEC is interesting, the rationale behind combining these three methods is not sufficiently articulated. For instance, why is BERT not directly integrated into the IDEC pipeline? Why use BTF-IDF on top of BERT embeddings instead of fine-tuning BERT or using sentence-level transformers like Sentence-BERT?
The mathematical description of BTF-IDF lacks rigor. Key variables (e.g., variance terms, α and β factors) are introduced with minimal notation explanation. It’s unclear how the scale penalty β is derived or normalized. The manuscript would benefit from a clearer formulation of each term and its computational role.
Although BTF-IDF is claimed to outperform traditional TF-IDF, no quantitative comparison is provided in the experiments. An ablation study with TF-IDF vs. BTF-IDF (both feeding into IDEC) would be necessary to isolate the benefit of this enhancement.

Validity of the findings

The paper lacks details on the architecture of the encoder/decoder used in IDEC. How many layers, activation functions, embedding size, and learning rate were used? How was the loss balancing parameter λ selected in Eq. (12)? These are essential for reproducibility and evaluating the novelty of the clustering process.
It appears the BERT model is used only for static feature extraction. Given the recent success of fine-tuned models in downstream NLP tasks, the authors should clarify whether BERT is fine-tuned on the education domain corpus, or provide justification for not doing so.
The IDEC structure involves joint optimization of reconstruction and clustering loss, but no visualization or diagram of the self-encoder path is given in Figure 1. A block diagram showing encoder-decoder components and how clustering is integrated would improve comprehension.
Section 3.4 outlines the ATT-Bi-GRU sentiment classifier in a brief pseudocode format, but lacks details such as attention head configuration, GRU layer size, dropout usage, and optimizer settings. Consider expanding this into a proper algorithm box or table.

Additional comments

There is no mention of how long training takes, or how BIF-DCN performs in terms of memory and runtime compared to simpler baselines like TF-IDF+KMeans. This is especially important for practical deployment on real-time feedback systems.
There is no mention of cross-validation, grid/random search, or evaluation on a validation set for tuning model parameters (e.g., number of clusters in IDEC, learning rate in ATT-Bi-GRU). This may raise concerns about overfitting to test data.

---

## Round 0.2 · accepted · Accept

Both reviewers have confirmed that the authors have addressed all of their comments.

·

Basic reporting

no comment

Experimental design

no comment

Validity of the findings

no comment

Additional comments

no comment

Reviewer 2 ·

Basic reporting

The author has accommodated all suggested reviews. Recommended for publication in current form.

Experimental design

NA

Validity of the findings

NA

Additional comments

Recommended for Acceptance